# Simulating the effect of evaluation unit size on eligibility to stop mass drug administration for lymphatic filariasis in Haiti

**Natalya Kostandova[1¤a], Luccene Desir[2¤b], Abdel Direny[3¤c], Alaine Knipes[4], Jean Frantz Lemoine[5], Carl Renand Fayette[6], Amy Kirby[1¤d], Katherine Gass[7]***

**1** Rollins School of Public Health, Emory University, Atlanta, Georgia, United States of America, **2** Hopital Ste. Croix, Haiti; University of Notre Dame, Notre Dame, Indiana, United States of America, **3** ENVISION Project, RTI International, Washington DC, United States of America, **4** Division of Parasitic Diseases and Malaria, Centers for Disease Control and Prevention, Atlanta, Georgia, United States of America, **5** Ministry of Public Health and Population, Port-au-Prince, Haiti, **6** IMA World Health, Port-au-Prince, Haiti, **7** NTD Support Center, Task Force for Global Health, Decatur, Georgia, United States of America

¤a Current address: Bloomberg School of Public Health, Johns Hopkins University, Baltimore, Maryland, United States of America
¤b Current address: Hispaniola Initiative, The Carter Center, Atlanta, Georgia, United States of America
¤c Current address: IMA World Health
¤d Current address: Waterborne Disease Prevention Branch, Centers for Disease Control and Prevention, Atlanta, Georgia, United States of America
* kgass@taskforce.org

**Data Availability Statement:** The data required to replicate this are being published in the COR-NTD Dataverse, available at: https://dataverse.unc.edu/

## Abstract

### Background

The Transmission Assessment Survey (TAS) is a decision-making tool to determine when transmission of lymphatic filariasis is presumed to have reached a level low enough that it cannot be sustained even in the absence of mass drug administration. The survey is applied over geographic areas, called evaluation units (EUs); existing World Health Organization guidelines limit EU size to a population of no more than 2 million people.

### Methodology/Principal findings

In 2015, TASs were conducted in 14 small EUs in Haiti. Simulations, using the observed TAS results, were performed to understand the potential programmatic impact had Haiti chosen to form larger EUs. Nine "combination-EUs" were formed by grouping adjacent EUs, and bootstrapping was used to simulate the expected TAS results.

When the combination-EUs were comprised of at least one "passing" and one "failing" EU, the majority of these combination-EU would pass the TAS 79% - 100% of the time. Even in the case when both component EUs had failed, the combination-EU was expected to "pass" 11% of the time.

Simulations of mini-TAS, a strategy with smaller power and hence smaller sample size than TAS, resulted in more conservative "passing" and "failing" when implemented in original EUs.

dataset.xhtml?persistentId=doi:10.15139/S3/
JUUSHC.

**Funding:** This work received financial support from
the Coalition for Operational Research on
Neglected Tropical Diseases, which is funded at
The Task Force for Global Health primarily by the
Bill & Melinda Gates Foundation (OPP1053230), by
the United States Agency for International
Development through its Neglected Tropical
Diseases Program, and with UK aid from the
British people. The funders had no role in study
design, data collection and analysis, decision to
publish, or preparation of the manuscript.

**Competing interests:** The authors have declared
that no competing interests exist.

## Conclusions/Significance

Our results demonstrate the high potential for misclassification when the average prevalence of lymphatic filariasis in the combined areas differs with regards to the TAS threshold. Of particular concern is the risk of "passing" larger EUs that include focal areas where prevalence is high enough to be potentially self-sustaining. Our results reaffirm the approach that Haiti took in forming smaller EUs. Where baseline or monitoring data show a high or heterogeneous prevalence, programs should leverage alternative strategies like mini-TAS in smaller EUs, or consider gathering additional data through spot check sites to advise EU formation.

### Author summary

Lymphatic filariasis is a disease caused by roundworms that may lead to disability, psychological problems, stigma, and lowered quality of life. One of the key strategies to control and eliminate lymphatic filariasis is mass drug administration (MDA), or repeated treatment of all at-risk people living in affected areas with an annual dose of medicine. To determine whether MDA can be stopped in a particular area, a transmission assessment survey (TAS) is conducted whereby a sample of children are tested for filarial antigen and proportion with a positive result is compared against a target threshold. Existing guidelines for delimiting the geographic areas to conduct TAS permit large evaluation units. In 2015, TASs were conducted in Haiti using more stringent criteria for forming evaluation units, resulting in much smaller geographic areas for evaluation. Using simulations, the authors found that, had Haiti followed the existing guidelines and assessed larger geographic areas, many of the areas might have been misclassified and MDA stopped prematurely in some settings. This research suggests that caution is needed when forming evaluation units for TAS, especially if the prevalence of lymphatic filariasis is not uniform.

## Introduction

Lymphatic filariasis (LF) is a vector-borne disease caused by nematodes, or roundworms, that reside in lymphatic vessels and can lead to debilitating disability, as well as stigma, psychological problems, and lowered quality of life [1,2]. The cornerstone of the global LF program is prevention through Mass Drug Administration (MDA). The primary objective of MDA is to lower the level of microfilaraemia in infected people so that, even after MDA is stopped, transmission cannot continue [3]. The World Health Organization recommends annual MDA to all those living in areas at risk until transmission is no longer deemed to be ongoing. Of the 72 countries considered endemic for lymphatic filariasis, 50 are considered to require MDA, of which only three have yet to start MDA; 17 countries have been validated as having eliminated LF as a public health problem [4].

There are costs associated with implementing MDA; consequently, to maximize the use of scarce public health resources, it is important for programs to know when MDA can be stopped with minimal risk of recrudescence. A 2011 study of communes in Haiti that received MDA found the cost of MDA distribution in the first year of the national strategic plan in just nine out of 55 communes to be $264,970. Extending this cost to all of the communes in program amounts to about $1,214,102 for just one year, not including the cost of albendazole [5].

In 2011, the World Health Organization (WHO) developed guidelines for determining when MDA can be stopped [3]. The geographic area across on which a decision to stop MDA will be based is called an evaluation unit (EU), and is often made up of a combination of MDA implementation units (IUs). An EU should not exceed two million people [3]. An EU should be comprised of epidemiologically homogeneous areas that have received at least five rounds of MDA, with at least 65% of the population swallowing the drugs each round, and the prevalence of circulating filarial antigen (CFA) in all sentinel and spot-check sites in an EU must be less than 2% [3]. If all of these conditions are satisfied, a Transmission Assessment Survey (TAS) is carried out to determine whether MDA should be stopped [3].

The target population for TAS is children 6 to 7 years old. In areas where over 75% of children are enrolled in primary schools, school-based surveys can be used for TAS, whereas community-based surveys are required in areas with lower school enrollment [3]. The tests and critical thresholds used to determine if an EU can safely stop treatment differ based on the type of LF and its vector. In areas where *Wuchereria bancrofti* is the endemic parasite, and the mosquito vector is *Culex* or *Anopheles*, decision rule and critical cut-off are set to determine if the upper one-sided 95% confidence limit around the CFA prevalence is less than 2% in order for the EU to 'pass' the TAS and safely stop MDA.

TAS is an example of a modified Lot Quality Assurance Sampling method, with schools or communities serving as the primary sampling unit (PSU). When the total number of PSUs in the EU is small (e.g., <40), PSUs are selected via systematic sampling, while cluster sampling is used in larger EUs. The TAS guidelines provide a table, which takes into account the total population of 6 to 7 year olds in the EU, the sampling methodology, and anticipated design effect, to determine the recommended sample size and critical cutoff value for the survey [3]. Upon completion of the survey, the observed number of positive tests is compared to a critical cutoff, designed to measure the target threshold with known error. In the case of the TAS, the critical cutoff is designed to measure a threshold of 2% (1% where *Aedes* is the vector), with <5% chance of Type I error (falsely rejecting the null hypothesis that the prevalence is above the target threshold) and maintaining power of at least 75% when the true prevalence is less than half the threshold. Practically, if the observed number of positive cases in a TAS is greater than the critical cutoff, the EU 'fails' and continues MDA for at least two more rounds; if the observed number of positive cases is less than or equal to the cutoff, the EU is considered to 'pass,' and can stop MDA [3].

Haiti is one of four countries in the Americas endemic for LF, bearing 90% of LF disease burden in the region. The species endemic to Haiti is *Wuchereria bancrofti* and the primary vector is the *Culex quinquefasciatus* mosquito [6]. In 2001, the CFA prevalence among children aged 6 to 11 was between 0 and 45%, with over 88% of all communes showing prevalence greater than 1% and thus qualifying for MDA according to WHO guidance [3]. In 2000, with support from the Ministry of Public Health and the Population (MSPP), the National Program to Eliminate LF (NPELF) was started. Despite hurricanes, a devastating earthquake, and a cholera outbreak, by 2012, NPELF was able to implement MDA nationwide, reaching more than eight million people, with estimated coverage of 71% [7]. By 2019, 122 of the 140 communes in Haiti passed at least one TAS and no longer required MDA [8].

Despite the tremendous success of the TAS at enabling over a thousand EUs to stop MDA for the global LF program, some evidence suggests that the TAS, as it is currently designed, may not be an effective tool for stopping MDA in all settings [9]. The focality of LF infection, which increases as transmission is driven towards elimination, calls the liberal size allowance (up to two million population) for EUs into question. For example, the epidemiology and geographic distribution of LF is likely to be very different for people living in a densely populated area with homogeneous vector distribution, as opposed to those living in a sparsely populated

area with varying altitudes, humidity, and vector distribution. As the heterogeneity of transmission increases, the ability of cluster surveys, such as the TAS, to capture the underlying variation diminishes and the likelihood that pockets of ongoing transmission will be missed is increased [10]. It is important to note that the current TAS guidance suggests grouping IUs is appropriate when they share similar epidemiological features; however, this advice does not seem to be universally followed by country programs.

Although reducing the size of an EU may improve the chances of including pockets with persistent transmission of LF if they exist, reducing the size of an EU, and thus increasing the number of EUs overall, would increase costs dramatically. The mean cost of a community-based TAS, based on a 2013 study in 13 countries, is $38,513, whereas the average cost of a school-based TAS is $18,239 [11]. Given the limited resources available to LF elimination programs, the guidelines for EU size should balance good decision-making with programmatic feasibility. At the same time, the additional costs of TAS in smaller EUs should be weighed against the costs of additional rounds of MDA, as well as the costs of misclassifying EUs.

In this study, TAS data from Haiti were used to perform simulations to explore the programmatic implications of EU size. In particular, the effect of using larger EUs for classifying an area as ready (or not) to stop MDA was explored by combining adjacent smaller EUs. In addition, the potential of using a TAS with a reduced sample size, referred to as a 'mini-TAS', in smaller EUs was considered as a potential cost-saving approach.

## Methods

### Ethical statement

Ethical clearance was not required for this study, as it was a secondary analysis of programmatic data. No personally identifying individual-level data were used in this analysis.

### Dataset

The dataset utilized in this study was a subset of data from a TAS-Soil-Transmitted Helminthiasis-Malaria survey conducted by the Haitian MSPP, IMA World Health, and Centers for Disease Control and Prevention (CDC) in 2015 in Haiti. The TAS was conducted in 14 EUs, with each unit comprised of one or more communes, third-level administrative divisions in Haiti, with the exception of one evaluation unit that was smaller than a commune. All EUs had completed the TAS eligibility requirements as established by WHO: at least 5 consecutive rounds of MDA with coverage over 65%; CFA prevalence at sentinel and spot-check sites of <2%; and a total population under two million people. The TAS were conducted using either a randomized cluster or systematic survey design targeting children 6–7 years old, with schools as the primary sampling unit. Immunochromatographic card test (ICT) was used to test for the presence of filarial antigens. The data collected included the names of each EU, the names and locations for each school, the ages and sex of the children tested, and the ICT results (positive, negative, indeterminate, and not available). Information from the Survey Sample Builder (http://www.ntdsupport.org/resources/transmission-assessment-survey-sample-builder) files for each EU was used to obtain information about the target population, total number of schools, and expected absentee rates for each EU. Henceforth these data will be referred to as the 'observed' data.

### Forming combo-EUs

In order to explore the implications of EU size, and because in Haiti EUs tend to have substantially fewer than two million people, larger EUs were simulated by combining adjacent EUs. In

this manner, nine unique combinations of adjacent EUs (hereby referred to as 'combo-EUs') were formed. Each of these new combo-EUs represented an alternative EU that the NPELF could have designated as the basis for its stopping MDA decision, as the combo-EUs would satisfy the TAS eligibility guidelines specified by WHO. Homogeneity criterion was not considered in forming combo-EUs, as baseline prevalence estimates were several years old, and becomes some other countries and TAS disregard homogeneity criterion when forming EUs. Target populations for each combo-EU were determined by combining the target populations for each component EU contained in the combo-EU. The total number of schools in the combo-EU was taken to be the sum of schools in each component EU. The expected absentee rate for each individual evaluation unit varied from 10% to 15%; since each of the combo-EUs contained at least one EU with an expected absentee rate of 15%, all of the combo-EUs were assigned the expected absentee rate of 15%. Because the target population of each of the combo-EUs exceeded 1000 and the number of schools in each combination exceeded 40, cluster sampling was assumed, as recommended by the WHO TAS guidelines. The WHO TAS table was used to obtain the necessary TAS sample size for the combo-EUs [3]. The average number of students per school was estimated by dividing the total target population of the combo-EU by the number of schools in the combo-EU. Finally, the target TAS sample size was divided by this average number of students to obtain the number of schools that needed to be sampled for each combo-EU, with a minimum of 30 schools required. If the sample size was not reached, additional children were sampled from a list of backup schools, selected proportionately from the EUs comprising the combo-EU.

## Passing or failing decision

In this study it was assumed that the programmatic decision for a combo-EU was to 'pass' the TAS if all component EUs passed the TAS (i.e., with the number of positive tests less than or equal to the critical cutoff), allowing MDA to be stopped. Whereas if any of the component EUs failed, the programmatic decision for the combo-EU was to fail, a conservative decision to avoid prematurely stopping MDA in areas with ongoing transmission.

A TAS in each combo-EU was treated as a stratified cluster survey, with component EUs acting as strata and schools as clusters. Sampling weights were assigned to each child with a positive or negative ICT, with the weights for children in EU $j$ defined as follows:

$$w_j = \frac{N_j}{n_j} \tag{1}$$

where $N_j$ is the target population in EU $j$ and $n_j$ is the number of children with a valid (positive or negative) ICT in the sample in EU $j$. The expected prevalence for the combo-EU was then obtained as a weighted average of each component EU's prevalence.

To assess the TAS critical cutoff, an upper one-sided 95% confidence interval was calculated for each expected prevalence accounting for the stratified cluster sampling using R package *survey*. If the confidence interval around the expected prevalence in the combo-EU contained or exceeded the TAS threshold of 2%, then the expected decision for the combo-EU was to fail; otherwise, the expected decision for the combo-EU was to pass.

## Bootstrapping

To understand the distribution of TAS results that one might expect had larger EUs been formed, bootstrapping, that is sampling with replacement from the observed data, was used to estimate the number of ICT positives if TAS were conducted in each combo-EU. In the first step, the estimated number of schools required to meet the TAS sample size for a combo-EU

was sampled with replacement from among all the schools in the observed TAS datasets for each of the component EUs. School selection was stratified by EU and schools were boot-strapped independently from each EU, with the number of selected schools proportional to the total number of schools in the EU. For those component EUs that were originally sampled systematically, rather than through cluster sampling, additional bootstrapping of children within the school was performed in order to obtain the necessary sample size. In these schools, the number of children selected was equal to the average number of children per school in the combo-EU. For EUs with cluster sampling, bootstrapping was only done at the school level, and results from all children that had been tested were retained. In some replicates, by chance, a disproportionate number of smaller schools was selected. As a result, the sample size was smaller than the target. In this case, if the target sample size of children was not reached from the schools selected through bootstrapping, additional schools were sampled until the desired sample size was met. This is consistent with how TAS is performed in the field, whereby additional randomly selected clusters are added if the target sample size is not met from the original sample of clusters. This bootstrap sampling was replicated 1000 times for each combo-EU, resulting in 1000 simulated TAS results. The total number of positive ICT results in each of the bootstrap replicates was calculated based on the number of ICT positive results in the observed TAS data for each selected school, and an upper 95% one-sided confidence interval was calculated for the combo-EU. If the confidence interval contained 2%, then the combo-EU was said to have failed; otherwise, the combo-EU passed. The proportion of replicates with upper one-sided 95% confidence intervals exceeding 2% was calculated.

It was necessary to drop EU #1 from the bootstrap simulations because an error in the original dataset, whereby schools 1 through 16 were all coded as "1," made it impossible to recreate the school-level results. A table with assessment of reproducibility of TAS results for component EUs using bootstrap is presented in Supporting Information (S2 Table).

## Mini-TAS

The alternative to combining IUs into EUs would be for each IU to be its own EU, a decision that comes with significant cost implications due to the increase in the number of TASs that would be required. Although the Haitian program chose to adopt this strategy, other programs might find it difficult to assume this added cost up front. The 'mini-TAS' represents a modification to the TAS platform that can reduce the cost and other resources required while still maintaining its integrity as a decision-making tool for stopping MDA. Simulations were run to compare the trade-offs of using the mini-TAS, in place of the TAS, for making stop-MDA decisions when each IU represents its own EU.

The mini-TAS is similar in design to the standard TAS. It is a 30-cluster survey designed to measure a threshold of 2% but requires testing roughly a quarter of the number of children of a standard TAS. This reduction in sample size, intended to reduce the time and cost associated with conducting a TAS, effectively reduces the power of the survey tool from 75% to 40%. The mini-TAS has been approved by WHO as a tool for confirmatory mapping of LF [12], and the details of its design have been well-documented [13]. The implications of conducting the mini-TAS were simulated in each EU in the observed Haiti dataset. The required sample size for the mini-TAS was based on the hypergeometric distribution so that each EU has no more than a 5% chance of being misclassified as passing when the true prevalence exceeds 2% (Type I error), and at least a 40% chance of correctly passing if the CFA prevalence is 1.0% (S1 Table). The bootstrapping approach was repeated as before, with replicates forced to achieve the desired sample size every time. For systematically sampled EUs, the number of children to sample from each school was calculated by multiplying the total mini-TAS sample size by the

proportion of valid ICT results in the school. If this sample size was not reached, additional children were sampled at random until the desired sample size was reached. For cluster surveys, the original mini-TAS design uses population proportionate to estimated size sampling to select the school clusters. To achieve an equal probability of selection, it is therefore necessary to use a cluster-specific sampling interval that is inversely proportional to the estimated size of the school. This results in a fixed expected sample size across all schools (which reduces to: per school sample size = total sample size / 30 clusters). To simulate this, at each school, the per school sample size was first drawn without replacement; if the original dataset had less than this required number of children with valid ICT results within the school, additional children were sampled with replacement from that school until the required number was reached. The number of passing and failing replicates out of the 1000 total replicates obtained for each EU was calculated in a similar manner to the TAS simulations. Upper one-sided 95% CI was calculated for each replicate, and the replicates were said to "pass" if the upper bound was less than 2%, and to "fail" if the upper bound was greater than or equal to 2%.

All analyses were conducted in R [14]. The package *survey* [15] was used for calculation of upper one-sided 95% confidence bound to allow for complex survey methodology.

## Results

### The TAS dataset

Information pertaining to characteristics of the EUs and TAS results from the observed data is presented in Table 1. Fourteen total EUs were sampled in TAS, with number of children in

**Table 1. Characteristics of individual Evaluation Units and Transmission Assessment Survey results.**

| Evaluation Unit # | Baseline prevalence of infection | Target population | Total schools in Evaluation Unit | Average # of students in target grades | Expected absentee rate | # Schools tested | Type of survey | # Children Tested | # Positive Results | Critical Cutoff | Observed Transmission Assessment Survey Decision |
|---|---|---|---|---|---|---|---|---|---|---|---|
| 1* | Low | 14,813 | 367 | 40 | 10% | 36 | Cluster | 1494 | 0 | 16 | Pass |
| 2* | Low | 35,357 | 721 | 49 | 10% | 46 | Cluster | 1659 | 3 | 18 | Pass |
| 3 | High | 2,442 | 67 | 36 | 10% | 53 | Cluster | 1231 | 2 | 14 | Pass |
| 4 | Medium | 6,821 | 120 | 57 | 10% | 45 | Cluster | 1528 | 0 | 18 | Pass |
| 5 | High | 707 | 17 | 42 | 10% | 16 | Systematic | 364 | 1 | 3 | Pass |
| 6 | Low | 18,977 | 333 | 57 | 10% | 42 | Cluster | 1617 | 2 | 18 | Pass |
| 7* | High | 1,597 | 25 | 64 | 15% | 25 | Systematic | 551 | 0 | 6 | Pass |
| 8* | Low | 20,833 | 441 | 47 | 15% | 47 | Cluster | 1587 | 2 | 18 | Pass |
| 9 | High | 754 | 26 | 29 | 15% | 24 | Systematic | 587 | 0 | 6 | Pass |
| 10 | High | 1,875 | 36 | 52 | 15% | 30 | Systematic | 672 | 0 | 7 | Pass |
| 11 | High | 1,336 | 42 | 32 | 15% | 31 | Cluster | 858 | 19 | 9 | Fail |
| 12 | High | 1,634 | 48 | 34 | 15% | 37 | Cluster | 1037 | 15 | 11 | Fail |
| 13 | High | 9,299 | 199 | 47 | 15% | 32 | Cluster | 1984 | 19 | 20 | Pass |
| 14 | High | 4,038 | 74 | 55 | 15% | 33 | Cluster | 1414 | 10 | 16 | Pass |

Baseline prevalence of infection is based on estimates from 2001 [16]. Evaluation Units (EUs) with Immunochromatographic card test (ICT) positivity between 0.1 and 4.9% are classified as low baseline prevalence; those with 5–9.9% ICT positivity have medium prevalence, and those with 10% and higher positivity are high prevalence at baseline. Target population is the expected number of school children enrolled in 1st and 2nd grades of primary schools. Number of schools in EU denotes the number of schools that exist in the evaluation unit. Number of schools tested is the number of schools that were selected in TAS, and for whom there is at least one ICT results present in the data. Number of children tested is the number of positive and negative ICT results that were recorded in the EU. If the number of positive ICT results in the EU is greater than the critical cutoff, the EU is said to fail; else, the EU passes. EUs marked with asterisks (*) were not considered for formation of combination-EUs because the combination-EU comprised of these adjacent units would have had a small enough number of positive results that failing would have been highly unlikely.

target grades in schools in the EUs ranging from 707 children 6–7 years to 35,357. Four of these EUs had low baseline prevalence of infection (0.1–4.9% ICT positivity), one had medium baseline prevalence (5.0–9.9% ICT positivity), and nine had high baseline prevalence of infection (10.0% and over ICT positivity) based on estimates from 2001 [16]. The number of schools in the EUs ranged from 17 to 721 and the average number of students in target grades per school ranged from 29 to 64.

The number of schools visited per EU as part of the TAS spanned from 16 in EU #7 to 53 schools in EU #3. Four of the EUs had <40 schools and required systematic sampling, meaning all schools that were accessible were sampled. The remaining ten EUs were sampled through cluster surveys, with the number of schools visited ranging from 31 to 53. In the EUs where cluster surveys were conducted, all children in the target grades were tested for CFA using the ICT test, whereas in systematically sampled EUs, a set fraction of students in the target grades were tested. The total number of children tested per EU ranged from 364 in EU #5 to 1986 in EU #13. Distribution of positive ICT results per school within EUs is provided in Supporting Information, S3 Table.

Two of the EUs, EU #11 and EU #12, failed the TAS, that is, the number of positive ICT results exceeded the critical cutoff. EU #13 passed the TAS but came close to reaching the critical cutoff, with 19 positive ICT results, compared to a cutoff of 20. All other EUs passed the TAS, with the number of positive ICT results far below the critical cutoff.

The EUs and the locations of schools where the surveys were conducted are displayed in Fig 1.

## Forming combo-EUs

Of the potential combo-EUs, those that were comprised solely of EUs with no positive ICT results, or an extremely small number of positive results (3 or less for EUs large enough to merit a cluster survey), such as EU #2 and EU #1, or EU #7 and EU #8, were not considered, as it would be expected that these combo-EUs result in a passing decision; their inclusion would

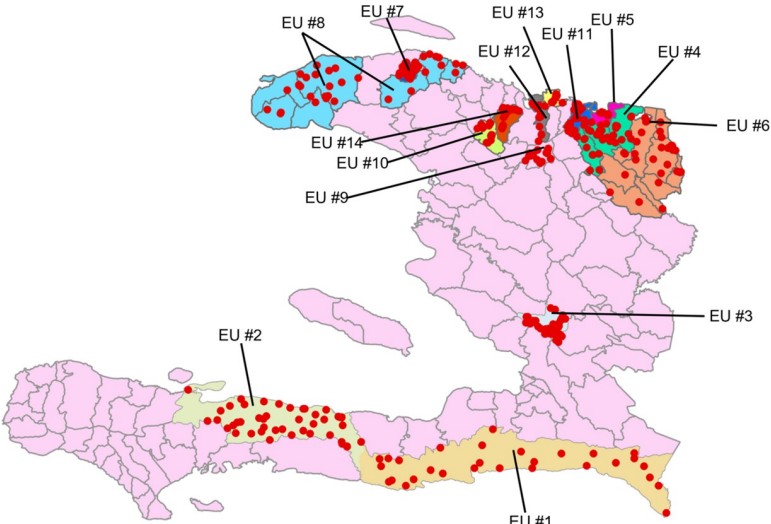

**Fig 1. Sites of Transmission Assessment Surveys and Evaluation Units.** Red circles represent schools where schoolchildren in grades 1 and 2 were tested. The administrative division shapefile that served as a base map is available at https://data.humdata.org/dataset/777e8b06-337f-4295-80bc-ca1515244215/resource/9b57a285-e12f-4d1a-b167-676d96a2b4af/download/hti_adm_cnigs_20181129.zip; the shapefile with Evaluation Unit number as an attribute is available for download at https://doi.org/10.15139/S3/JUUSHC.

**Table 2. Characteristics of combination Evaluation Units, formed from adjoining Evaluation Units.**

| Evaluation Unit Combination | Component Evaluation Units | Observed Decision | Target sample size | # Schools to be sampled | Programmatic decision | Expected true prevalence (upper 1-sided Confidence Interval) | Expected transmission assessment conclusion |
|---|---|---|---|---|---|---|---|
| A | 12 | Fail | 1540 | 41 | Fail | 1.03% (1.58%) | Pass |
|   | 13 | Pass |  |  |  |  |  |
| B | 12 | Fail | 909 | 33 | Fail | 0.99% (2.11%) | Fail |
|   | 9 | Pass |  |  |  |  |  |
| C | 12 | Fail | 1540 | 43 | Fail | 0.96% (1.48%) | Pass |
|   | 13 | Pass |  |  |  |  |  |
|   | 9 | Pass |  |  |  |  |  |
| D | 11 | Fail | 909 | 31 | Fail | 1.54% (2.70%) | Fail |
|   | 5 | Pass |  |  |  |  |  |
| E | 11 | Fail | 1532 | 36 | Fail | 0.36% (0.61%) | Pass |
|   | 4 | Pass |  |  |  |  |  |
|   | 5 | Pass |  |  |  |  |  |
| F | 11 | Fail | 1556 | 34 | Fail | 0.20% (0.36%) | Pass |
|   | 4 | Pass |  |  |  |  |  |
|   | 5 | Pass |  |  |  |  |  |
|   | 6 | Pass |  |  |  |  |  |
| G | 10 | Pass | 1392 | 31 | Pass | 0.48% (0.83%) | Pass |
|   | 14 | Pass |  |  |  |  |  |
| H | 11 | Fail | 1556 | 34 | Fail | 0.20% (0.36%) | Pass |
|   | 4 | Pass |  |  |  |  |  |
|   | 6 | Pass |  |  |  |  |  |
| I | 11 | Fail | 1356 | 49 | Fail | 1.79% (2.80%) | Fail |
|   | 12 | Fail |  |  |  |  |  |

Positive Immunochromatographic card tests (ICTs), Critical Cutoff, Decision, and # schools tested all refer to individual characteristics of the component Evaluation Units (EUs) that make up the combination EUs (combo-EUs). Target sample size is the number of children that should be selected via bootstrapping to achieve desired power and alpha levels. Number of schools sampled is the expected number of schools (aka clusters) that will need to be selected from the combo-EU in order to achieve the desired sample size, sampled proportionately to total number of schools in the component EUs. Programmatic decision is to fail if at least one of the individual EUs is said to fail; if all individual EUs comprising the combo-EU pass, the desired conclusion is to pass. The expected true prevalence is the weighted average of prevalence in the EUs comprising the combo-EU. The expected Transmission Assessment Survey decision is to fail the combo-EU if the upper one-sided 95% confidence interval of the expected true prevalence is greater than or equals 2%, and to pass otherwise.

not be informative. This left nine combo-EUs for the simulations; a description of these combo-EUs is presented in Table 2.

As seen in Table 2, the expected TAS decision, based on the expected prevalence of positive ICT results from the weighted average of the component EUs, differed from the programmatic decision for five out of the nine combo-EUs. That is, although the programmatic decision for the combo-EU was to fail if at least one of its component EUs had failed the TAS, in five of the combo-EUs that had at least one component EU that failed the TAS, the upper one-sided 95% CI around the expected prevalence was less than 2%, indicating a passing result. Thus, for these combo-EUs, there was a discordance between the desired and expected decisions.

## Combo-EU Bootstrapping

The results from the bootstrapping to obtain the distribution of likely TAS results for each combo-EU are shown in Table 3. When the combo-EUs were comprised of EUs with the same

**Table 3. Results of bootstrapping results simulating Transmission Assessment Surveys in combination Evaluation Units.**

| Evaluation Unit Combination | Programmatic decision | Median bootstrap prevalence (upper 1-sided 95% Confidence Interval) | Bootstrap expected conclusion | EU | # of schools selected from each EU | % of replicates failing Transmission Assessment Survey (out of 1,000) |
|---|---|---|---|---|---|---|
| A | Fail | 0.97% (1.42%) | Pass | 12 | 8 | 18.2% |
|   |      |               |      | 13 | 33 |       |
| B | Fail | 1.12% (2.08%) | Fail | 12 | 22 | 61.6% |
|   |      |               |      | 9  | 12 |       |
| C | Fail | 0.96% (1.46%) | Pass | 12 | 8  | 21.1% |
|   |      |               |      | 13 | 31 |       |
|   |      |               |      | 9  | 5  |       |
| D | Fail | 2.01% (3.41%) | Fail | 11 | 22 | 93.2% |
|   |      |               |      | 5  | 9  |       |
| E | Fail | 0.08% (0.28%) | Pass | 11 | 9  | 0.2% |
|   |      |               |      | 4  | 25 |       |
|   |      |               |      | 5  | 4  |       |
| F | Fail | 0.11% (0.28%) | Pass | 11 | 3  | 0.0% |
|   |      |               |      | 5  | 2  |       |
|   |      |               |      | 4  | 8  |       |
|   |      |               |      | 6  | 22 |       |
| G | Pass | 0.58% (0.95%) | Pass | 10 | 10 | 1.9% |
|   |      |               |      | 14 | 21 |       |
| H | Fail | 0.11% (0.27%) | Pass | 11 | 3  | 0.0% |
|   |      |               |      | 4  | 9  |       |
|   |      |               |      | 6  | 23 |       |
| I | Fail | 1.78% (2.52%) | Fail | 11 | 23 | 89.3% |
|   |      |               |      | 12 | 26 |       |

Replicates are obtained by proportional sampling. Programmatic decision is to fail if at least one of the individual Evaluation Units (EUs) is said to fail; if all individual EUs comprising the EU combination pass, the desired conclusion is to pass. The median bootstrap prevalence is the expected prevalence of positive Immunochromatographic card Test results in the bootstrap of 1000 replicated. The bootstrap expected conclusion is to fail the EU combination if the upper one-sided 95% confidence interval exceeds 2%, and to pass otherwise. The Number of baseline schools selected refers to the number of schools selected from each individual EU to be proportional to the total number of schools in the EU, relative to the number of schools in the EU combination. Additional schools were sampled if desired sample size was not achieved.

observed TAS decision–that is, with all component EUs failing, or all passing–the bootstrapping simulations produced the same decision in the majority of the replicates. In the case of combo-EU G, comprised of component EUs #10 & #14 that both passed TAS, 981 out of 1000 replicates also passed TAS (1.9% failed).

For the combo-EU I, comprised of two failing EUs (#11 & #12), the vast majority of bootstrap replicates (89.3%) also failed the TAS.

For the eight combo-EUs comprised of component EUs with discordant TAS decisions, the programmatic decision is for the combo-EU to fail the TAS. However, as seen in Table 3, the rate by which these combo-EUs failed the TAS was highly variable. Combo-EU D, comprised of EUs #11 and #5, and combo-EU B, comprised of EUs #12 and #9, had the highest percentage of failing replicates, with 93.2% and 61.6% of replicates failing TAS, respectively. For the remaining six combo-EUs comprised of EUs with discordant TAS results, the rate of TAS failure ranged from 0% in the case of combo-EUs F and H, to 21.1% for the combo-EU C.

**Table 4. Results of mini-Transmission Assessment Survey (mini-TAS) simulations.**

| Evaluation Unit # | Observed Transmission Assessment Survey Decision | Mini-Transmission Assessment Survey type | Mini-Transmission Assessment Survey Sample Size | Mini-Transmission Assessment Survey Critical Cutoff | % of replicates that fail mini-Transmission Assessment Survey |
|---|---|---|---|---|---|
| 2 | Pass | Cluster | 480 | 3 | 0.0% |
| 3 | Pass | Cluster | 480 | 3 | 1.1% |
| 4 | Pass | Cluster | 480 | 3 | 0.0% |
| 5 | Pass | Systematic | 220 | 1 | 0.0% |
| 6 | Pass | Cluster | 480 | 3 | 0.9% |
| 7 | Pass | Systematic | 300 | 2 | 0.0% |
| 8 | Pass | Cluster | 480 | 3 | 0.0% |
| 9 | Pass | Systematic | 220 | 1 | 0.0% |
| 10 | Pass | Systematic | 300 | 2 | 0.0% |
| 11 | Fail | Cluster | 450 | 3 | 100.0% |
| 12 | Fail | Cluster | 450 | 3 | 100.0% |
| 13 | Pass | Cluster | 480 | 3 | 30.4% |
| 14 | Pass | Cluster | 480 | 3 | 100.0% |

Mini-TAS mimics the TAS procedure, with power reduced to 40%, effectively reducing sample size. One thousand replicates are obtained through bootstrapping; replicates were declared to "pass" the mini-TAS if the number of positive Immunochromatographic card Test results in the replicate was less than or equal to the critical cutoff; otherwise, the replicate was considered to have failed the mini-TAS.

## Mini-TAS

The results of mini-TAS simulations are presented in Table 4. The vast majority of the mini-TAS bootstrap replicates passed the TAS. In seven of the thirteen EUs, all of the bootstrap replicates would pass in the mini-TAS, which is intuitive because the total number of positive ICTs in the full TAS sample was at or below the cut-off threshold for mini-TAS. In two other EUs, where the observed TAS decision was to pass, mini-TAS would have resulted in a failing decision a small portion of the time (1.1% for EU #3 and 0.9% for EU #6). For the EU with the borderline passing TAS decision, EU #13, mini-TAS would have failed 30% of the time. The two EUs that failed in TAS also failed 100% of the mini-TAS replicates. EU #14, on the other hand, failed 100% of mini-TAS replicates, despite having passed the TAS.

## Discussion

The TAS is a statistically robust decision-making tool that has been successfully implemented by program managers in many countries and used to guide important stop-MDA decisions for LF. While WHO provides strong guidance on how to conduct and interpret TASs, the best practices for forming survey evaluation units are vague, particularly when it comes to recommended EU size. In this study, programmatic data from Haiti's LF elimination program were used to simulate various EU formations and the resulting programmatic decisions regarding the decision to stop MDA. The study's results suggest that there is a high potential for misclassifying areas where MDA should not be stopped when such implementation areas are combined with low prevalence areas into a single EU. In fact, of the eight EU combinations for which the desired program conclusion was to fail the TAS, five EU combinations would be expected to pass at least 79% of the time. For all combo-EU replicates, the bootstrap expected decision conformed with the expected true prevalence decision based on a weighted average of prevalence of each of the comprising EUs. Unfortunately, this decision was different from the programmatic decision in the vast majority of the combinations, which would be to fail the

combo-EU if any of the comprising EUs should fail TAS. It should be noted that the two EUs that failed TAS (EU #11 and EU #12) had below average target population; when combined with larger EUs, which passed TAS, the probability of sampling a high enough number of the positive ICTs from EU #11 and EU #12 was low. Only when combining the two failing EUs with even smaller EUs with low prevalence were we more likely to fail the combination-EUs (eg combination-EU D).

The high rate of disagreement of results with both the expected decision and the desired decision is concerning. TAS is used to assess whether MDA for LF can be stopped. Falsely passing a combination EU in which one or more of the composite EUs should have failed could have significant public health consequences and jeopardize elimination efforts. With MDA prematurely stopped, transmission would continue unabated for at least two years before a second TAS could be carried out and the program would have a chance at recognizing the error. Once the error was identified, restarting an MDA program in an EU previously declared free of transmission would require significant human and financial resources and would incur a political capital cost. This study suggests that prematurely stopping MDA might be the more likely form of misclassification when IUs are combined, a concerning conclusion.

The financial and logistical challenges of conducting TAS are significant and thus the desire to combine IUs into a larger single EU to reduce that burden is understandable; however, it can be difficult to know which IUs are appropriate to combine. Although it might seem obvious that combining two IUs with discordant results (i.e., one pass and one fail) would lead to an incorrect decision for one of the component IUs, it is important to keep in mind that programs do not have this information in advance when they are determining whether to combine IUs. In its TAS manual, WHO advises that IUs can be combined if they have had at least five rounds of MDA and share "similar epidemiological features" [3]. The manual suggests that the epidemiological features of interest can include rates of MDA coverage and prevalence in sentinel and spot-check sites. Currently, the manual recommends that there be at least one sentinel site per one million population, with at least one corresponding spot-check site [3]. As seen in S1 Fig, which is an ArcGIS-generated map of the distribution of positive ICT results from TAS in the northern EUs, positive cases appear to cluster. Because of the focality of LF, particularly towards the end-stages of the program, as the size of the EU increases, so does the likelihood that the cluster sampling used in the TAS will miss a hotspot of ongoing transmission [10]. Although limiting the size of the EU is the best way to reduce the risk of undetected hotspots, an alternative strategy might be to increase the number of pre-TAS sentinel and spot-check sites prior to selection of EUs. If the pre-TAS data suggest some low level of infection remain (e.g., CFA between 1% and 2%), it might be prudent to restrict the corresponding IU to a single EU.

One method for addressing the tradeoff between the improved decision-making power that comes with smaller EU size vs. the added costs and resources that more EUs represent, is to use the mini-TAS, in place of the TAS. Because the mini-TAS sample size is much smaller, a single team can typically complete sampling in two clusters (i.e., schools) per day, which may result in two- or three-fold savings in survey implementation costs [13]. While cost effectiveness analysis of switching to mini-TAS approach is outside the scope of this study, published experiences with both tools in Tanzania suggest that the mini-TAS costs $9,598 per EU [13] while the cost of TAS is $29,721 [11]. Based on our analysis, using a mini-TAS would tend to provide more conservative results that favor continuing MDA compared with the TAS (a consequence of reducing the power from 75% down to 40%). In the nine high-performing EUs with zero or very few ICT positives during TAS, the simulations suggest that the mini-TAS would be likely to agree with the TAS and the EU would be classified as 'passing' >98% of the time (100% of the time for those with no positives, as expected, as well as three of the EUs with

a low number of positives). In the EUs that failed the TAS, it is reassuring to observe that they would likely fail the mini-TAS 100% of the time. In EUs where the TAS results were borderline (EUs #13 & 14), the mini-TAS was more likely to fail the EU compared to the TAS, failing 70% of the time for EU #13 and 100% of the time for EU #14. This might have occurred because out of 33 schools in this EU, eight had at least one positive ICT. With the low cut-off threshold in mini-TAS (three positive ICT), it is likely that cluster sampling would have picked up a high enough number of these positive results to trigger a failing decision. While some NTD practitioners might find this increase in failures concerning, others might argue that it is the more conservative decision particularly in light of recent evidence that the TAS might not be sufficiently sensitive for detecting ongoing transmission in all settings [9,17].

Although our study focused on the issue of combining IUs to form EUs, in some countries dense population and district structure might result in IUs that approach, or even exceed, two million population. In this case, the question is not about combining IUs but whether it makes sense to split IUs into smaller EUs when conducting the TAS. Here again it becomes an important trade-off between accurate decision-making and cost. Subdividing large IUs to form smaller EUs offers two advantages: 1) MDA can be stopped in the portions of the IU where treatment was successful and 2) reducing the area over which disease prevalence is being averaged decreases the risk that "transmission hotspots" go undetected [18]. Here too, leveraging the mini-TAS to make stop-MDA decisions in these smaller EUs might provide a strategy to maintain the robust design and decision-making power of the TAS, while reducing the overall cost and material requirements to the program.

It is important to note that the simulation approach taken here of directly combining data from two or more EUs may not be an appropriate way to estimate real-life TAS results. In particular, it was difficult to identify the most appropriate way to combine observed TAS data from EUs that used discordant sampling methods (systematic vs. cluster sampling). As with any bootstrap sampling approach, this analysis was limited to samples that had been obtained during TAS. Where the prevalence is heterogeneous, cluster-based surveys (such as the TAS) may miss small foci of infection by chance and these foci would not be reflected in the subsequent bootstrap simulations.

The results from these simulations suggest that epidemiological characteristics, rather than total population or geographic size, should be given the greatest consideration when forming EUs. Furthermore, these results suggest that the strategy adopted by the Haitian program to limit EUs to a single IU (i.e. commune) in areas where baseline transmission intensity was high was a wise and conservative approach that likely averted misclassification of EUs. This strategy makes sense, as areas with historically high transmission intensity are likely to be more vulnerable to recrudescence or harboring pockets of focal transmission. Cluster sample surveys, such as the TAS, are limited in their ability to detect focal transmission. Restricting the total size of areas at greatest risk increases the chance of detecting focal transmission and making the correct treatment decision. Ultimately, the decision of EU size is based on availability of good information and financial resources. Where baseline information is available, we recommend that it factor into the decision to combine IUs, in the case of low transmission settings, or keep separate, in the case of areas with historically high transmission. Providing a precise threshold to determine whether combining or splitting IUs is indicated is unrealistic, given the sparsity of most baseline data and the relevance of other epidemiologic factors. Programs must also consider the cost benefits of conducting fewer TAS evaluations with the increased risk of EU misclassification. The mini-TAS represents a potential compromise for programs, as it provides a strategy to maintain the robust design and decision-making power of the TAS, while reducing the overall cost and resource requirements. Ultimately program managers should continue to make thoughtful decisions when forming EUs to improve the

likelihood that appropriate stop-MDA decisions are made and enable programs to reach their elimination goals as efficiently as possible.

## Supporting information

**S1 Fig. Spatial distribution of positive Immunochromatographic card test results in Haiti Transmission.** The administrative division shapefile that served as a base map is available at https://data.humdata.org/dataset/777e8b06-337f-4295-80bc-ca1515244215/resource/9b57a285-e12f-4d1a-b167-676d96a2b4af/download/hti_adm_cnigs_20181129.zip; the shape-file with Evaluation Unit number as an attribute is available for download https://doi.org/10.15139/S3/JUUSHC. Assessment Survey data.
(TIF)

**S1 Table. Decision rules and sample size for mini-Transmission Assessment Surveys.** Table adapted from [13].
(PDF)

**S2 Table. Comparison of observed 2015 Haiti Transmission Assessment Survey results in 13 Evaluation Units and simulated results using bootstrapping.**
(PDF)

**S3 Table. Distribution of positive Immunochromatographic card test results within Evaluation Units.**
(PDF)

## Acknowledgments

The authors would like to acknowledge the work of the Haitian Ministry of Health and IMA World Health and the team members that supported the TAS. The authors would also like to thank Michael Deming, for his insights into TAS methodology and simulation design, Frank Monestime, for his insights to Haiti TAS, and Patrick Lammie, for his guidance on the global program priorities. The findings and conclusions in this report are those of the author(s) and do not necessarily represent the official position of the Centers for Disease Control and Prevention (CDC).

## Author Contributions

**Conceptualization:** Natalya Kostandova, Amy Kirby, Katherine Gass.

**Formal analysis:** Natalya Kostandova.

**Investigation:** Luccene Desir, Abdel Direny, Alaine Knipes, Jean Frantz Lemoine, Carl Renand Fayette.

**Methodology:** Natalya Kostandova, Amy Kirby, Katherine Gass.

**Software:** Natalya Kostandova.

**Supervision:** Katherine Gass.

**Visualization:** Natalya Kostandova, Amy Kirby, Katherine Gass.

**Writing – original draft:** Natalya Kostandova, Katherine Gass.

**Writing – review & editing:** Natalya Kostandova, Alaine Knipes, Katherine Gass.

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
