## [Decision Letter · Decision Letter 0]

1 Oct 2021

Dear Dr Gass,

Thank you very much for submitting your manuscript "Simulating the effect of evaluation unit size on eligibility to stop mass drug administration for lymphatic filariasis in Haiti" for consideration at PLOS Neglected Tropical Diseases. As with all papers reviewed by the journal, your manuscript was reviewed by members of the editorial board and by several independent reviewers. The reviewers appreciated the attention to an important topic. Based on the reviews, we are likely to accept this manuscript for publication, providing that you modify the manuscript according to the review recommendations. 

With the revised submission, please include the original data in a form that allows the results to be checked and replicated. The reference to an aggregate summary of the data provided in reference 12 is not sufficient. If there are concerns to make the original microdata available to readers, if possible, please aggregate the microdata to a level that allows the bootstrap analysis to be reproduced, e.g., in the form of the number of types of test result per school (positives, negatives, indeterminate). This would allow the authors to at least drop the age and sex-related data, which based on the provided information, did not seem to play a role in the bootstrap procedure. However, if age and sex did play a role in the bootstrap procedure, the microdata should be made available without aggregation.

I further encourage the authors to reconsider dropping EU #1 from the bootstrap simulations ("because an error in the original dataset made it impossible to recreate the school-level results"). Although the nature of the error is not fully clear, it sounds like a technical problem. Given that bootstrapping is a well-established technique, it should be possible to resolve a technical issue and/or find help to resolve it. Or, if the authors can provide more details about the problem in their response, perhaps the reviewers can provide a suggestion.

Sincerely,

Luc E. Coffeng, MD PhD

Guest Editor

Jennifer Keiser

Deputy Editor

With the revised submission, please include the original data in a form that allows the results to be checked and replicated. The reference to an aggregate summary of the data provided in reference 12 is not sufficient. If there are concerns to make the original microdata available to readers, if possible, please aggregate the microdata to a level that allows the bootstrap analysis to be reproduced, e.g., in the form of the number of types of test result per school (positives, negatives, indeterminate). This would allow the authors to at least drop the age and sex-related data, which based on the provided information, did not seem to play a role in the bootstrap procedure. However, if age and sex did play a role in the bootstrap procedure, the microdata should be made available without aggregation.

I further encourage the authors to reconsider dropping EU #1 from the bootstrap simulations ("because an error in the original dataset made it impossible to recreate the school-level results"). Although the nature of the error is not fully clear, it sounds like a technical problem. Given that bootstrapping is a well-established technique, it should be possible to resolve a technical issue and/or find help to resolve it. Or, if the authors can provide more details about the problem in their response, perhaps the reviewers can provide a suggestion.

Reviewer's Responses to Questions

**Key Review Criteria Required for Acceptance?**

**Methods**

-Are the objectives of the study clearly articulated with a clear testable hypothesis stated?

-Is the study design appropriate to address the stated objectives?

-Is the population clearly described and appropriate for the hypothesis being tested?

-Is the sample size sufficient to ensure adequate power to address the hypothesis being tested?

-Were correct statistical analysis used to support conclusions?

-Are there concerns about ethical or regulatory requirements being met?

Reviewer #1: Methods were clearly defined and appropriate.

Reviewer #2: (No Response)

Reviewer #3: The objectives of the study are clearly defined. The authors test whether reasonable combinations of evaluation units would yield different programmatic decisions. The data and methods are appropriate for the questions, and the decisions to exclude some of the observed data from the deeper analysis are reasonable. One recommendation would be to add clarity on how additional clusters or children from clusters were drawn for the bootstrapping analysis. It seems odd that an analysis using totally sufficient samples could result in too small of a sample using essentially the same design.

**Results**

-Does the analysis presented match the analysis plan?

-Are the results clearly and completely presented?

-Are the figures (Tables, Images) of sufficient quality for clarity?

Reviewer #1: General comment re: presentation of results: 

I know “Desired programmatic decision” is defined, however it seems odd to state that any desired programmatic decision is to “Fail” – and to see that written in Table 3 just seems odd. If other language could be used to describe the same phenomena, I recommend using different language here (and in the text) so the layperson doesn’t actually ever think a program would desire to fail. Perhaps just drop the word “Desired”? Or replace it with "Purported"?

Reviewer #2: (No Response)

Reviewer #3: The tables and figures are clear and informative. The results are well described.

**Conclusions**

-Are the conclusions supported by the data presented?

-Are the limitations of analysis clearly described?

-Do the authors discuss how these data can be helpful to advance our understanding of the topic under study?

-Is public health relevance addressed?

Reviewer #1: Conclusions and considerations for programmatic decisions are clearly stated.

Reviewer #2: (No Response)

Reviewer #3: The conclusions suit the results and the scope of the analysis. The authors clearly describe the limitations of their work and the underlying data. The relevance and applicability are well treated.

**Editorial and Data Presentation Modifications?**

Reviewer #1: Minor suggested revisions below:

Line 15, replace Hispaniola Program with Hispaniola Initiative

Line 91: Recommend moving sentence “The only guidance…” before the sentence that starts on line 88 “An EU should be comprised…”

Line 149: First use of STH – but hasn’t been spelled out previously.

Line 276: “Error” Reference source not found.” Is printed in lieueof reference.

Line 281: Missing word “of” between “number students”

Line 283: Recommend formatting Table 1 to emulate Tables 2 - 4 (e.g., same font size)

Line 300: Recommend adding the words “passed the TAS but” btwn “EU #13” and “came close to”

Line 304: Capitalize the “F” in “figure 1”

Line 306. Transmission Assessment Surveys and Evaluation Units could be capitalized – see Table 3 – or review all figure and table titles throughout and be consistent.

Line 337: Add “the” btwn “in majority”

Line 343: Everywhere else, “card test” is lowercase except here. Choose one and be consistent throughout.

Line 369: Everywhere else, “card test” is lowercase except here. Choose one and be consistent throughout.

Line 373: Consider replacing “stopping-treatment decisions” with “stopping-MDA decisions” or “stop-MDA decisions” as is used elsewhere in manuscript.

Line 374: Replace TAS surveys with “TASs” – which has been used elsewhere in manuscript.

Line 466: “stop MDA decisions was earlier hyphenated as “stop-MDA decisions” – review manuscript in entirety and be consistent.

Line 533: S1 Figure, spell out acronyms.

Reviewer #2: (No Response)

Reviewer #3: (1) The abstract does not mention the mini-TAS analysis. If there is room, please briefly note that you explored alternative designs as a compromise. 

(2) While the use of the upper 95% CI to determine the TAS result based on bootstrapping is functionally equivalent to the LQAS, most readers will be more familiar with decision rules and cutoffs when it comes to TAS data. It may be helpful to reiterate this in the results, that the simulations produced a frequency of positive ICTs above the cutoff.

(3) The authors note in line 256 that the mini-TAS uses "population proportionate to estimated size sampling, which results in a fixed sample size across all schools." Is this not the other way around, that the PPES sampling uses a fixed fraction rather than a fixed number?

(4) There is a reference error in line 276.

(5) Table 1: perhaps include a column noting which EUs will be excluded from further analysis due to low number of positive children.

(6) Around line 414-418, the authors note that "Because the mini-TAS sample size is much smaller, a single team can

417 typically complete sampling in two clusters (i.e., schools) per day, which may result in two- or three-fold

418 savings in survey implementation costs." With school-based sampling and good mobilization, two clusters per day is plausible in a regular TAS. What are the estimated savings in labor and supplies that come from the smaller number of clusters and overall smaller sample size of the mini-TAS when compared to a regular TAS, particularly one done on a tighter schedule?

(7) Lines 444-446: did you notice a difference in the results when drawing from discordant sample designs (cluster v. systematic)? Perhaps in the variance? Intuitively, it seems like the very different population structures would affect the results in some way.

(8) Lines 450-451: Would reducing the upper population threshold from 2 million to say 1 million or 500,000 make a difference? Please elaborate more on why epidemiological conditions (presumably, high baselines leading to persistent transmission after years of MDA) matter more than the other factors. Moreover, programs often only have baselines and perhaps an interim survey or two before Pre-TAS. Beyond increasing the frequency of interim monitoring or the number of sentinel and spot check sites, how should programs classify their baselines? I.e., does 5% go with 10%, does 10% go with 15%, etc. This may be an area of further work and another paper, but a rough +/- x% may be helpful.

**Summary and General Comments**

Reviewer #1: Very well-written with clear explanation of relevance to programmatic decision-making.

Reviewer #2: I have read the paper “Simulating the effect of evaluation unit size on eligibility to stop mass drug administration for lymphatic filariasis in Haiti” with great interest. It is a great piece of work with clear policy relevance for the global programme to eliminate lymphatic filariasis. I have some suggestions and minor comments

SUGGESTIONS:

1. I miss an assessment of the reproducibility of TAS results with current EU-size and guidelines. Could you do 1000 bootstraps of the data within each component EU to assess how often that EU would be classified as passing or failing TAS? 

2. Table 1: could you add information about the number of positives per school (e.g. no positives in x schools, 1 positive in y schools, etc), so that readers have complete information to reproduce the analyses?

3. Table 1: it is interesting to see how the number of children tested varies with population size. In the smallest EUs about half of the target population is being tested. This declines to about 5% in the largest EUs. Similarly, nearly all schools are surveyed in small EUs, while only 10% of schools is surveyed in the largest EUs. Explaining the rationale behind the survey sample builder is perhaps beyond the scope of this paper, but it might be useful for readers to be reminded about this and to understand under which circumstances this would be appropriate. 

MINOR COMMENTS

4. Line 123: I’d suggest to delete the word “safely”, as the appropriateness of the decision often remains to be seen

5. Lines 123-133: this text is written as if there was not any statement about homogeneity in WHO’s guidance for creating EUs. But there was a statement about this in the guidelines. Perhaps one or two sentences about why this has not always been effectively applied would be helpful in the stage. Was there a lack of guidance on how to define whether an area is sufficiently homogenous to be considered one EU? 

6. Line 155-157: was the choice of EU-sizes in Haiti driven by information on homogeneity?

7. Line 171-173: state explicitly that the homogeneity criterion was not considered in forming combo-EUs

8. Line 291-292: EU numbers don’t match to the numbers in table 1.

9. Figure 1: the red is difficult to see (many points appear black to me, possibly because of the black border)

10. Line 353: did you intend to also mention the results for combo-EU B in this sentence (38.4%)? Replace “failing” by “passing”

11. Lines 354-356: it is a bit confusing that passing rates are provided in table 3 and that failing rates are discussed in the text. I suggest to harmonize this. In the discussion of the results presented in table 3, it may be useful to point out that target population was relatively small in the two component EUs that failed TAS in the observed data. When these EUs are combined with considerably larger EUs with low baseline prevalence, these component EUs make up a minor part of the total population. Only when combined with even smaller EUs, we still get a signal.

12. Lines 397-399: I don’t understand the first part of this sentence. How does combining IUs into larger EUs prevent “misclassifying well-performing EUs as failing”? Even with small EUs they should pass the criteria.

Reviewer #3: This paper is relevant to the NTD community. The methods are appropriate and straightforward. It is well written and clearly presented.

PLOS authors have the option to publish the peer review history of their article (what does this mean?). If published, this will include your full peer review and any attached files.

Reviewer #1: No

Reviewer #2: No

Reviewer #3: No

Figure Files:

Data Requirements:

Reproducibility:

References

---

## [Decision Letter · Decision Letter 1]

21 Dec 2021

Dear Dr Gass,

Thank you very much for submitting your manuscript "Simulating the effect of evaluation unit size on eligibility to stop mass drug administration for lymphatic filariasis in Haiti" for consideration at PLOS Neglected Tropical Diseases. As with all papers reviewed by the journal, your manuscript was reviewed by members of the editorial board and by several independent reviewers. The reviewers appreciated the attention to an important topic. Based on the reviews, we are likely to accept this manuscript for publication, providing that you modify the manuscript according to the review recommendations. 

The reviewers and editor appreciate the thorough reply and revisions. To be able to accept the manuscript for publication, I ask you to please check and address the last two points about line 198 and table 3.

Sincerely,

Luc E. Coffeng, MD PhD

Guest Editor

Jennifer Keiser

Deputy Editor

The reviewers and editor appreciate the thorough reply and revisions. To be able to accept the manuscript for publication, I ask you to please check and address the last two points about line 198 and table 3.

Reviewer's Responses to Questions

**Summary and General Comments**

Reviewer #2: The authors have adequately adressed my comments on the previous version and I have no further comments.

Reviewer #3: Great work, and thank you for your clear and thorough response.

PLOS authors have the option to publish the peer review history of their article (what does this mean?). If published, this will include your full peer review and any attached files.

Reviewer #1: No

Reviewer #2: No

Reviewer #3: No

**Key Review Criteria Required for Acceptance?**

**Methods**

-Are the objectives of the study clearly articulated with a clear testable hypothesis stated?

-Is the study design appropriate to address the stated objectives?

-Is the population clearly described and appropriate for the hypothesis being tested?

-Is the sample size sufficient to ensure adequate power to address the hypothesis being tested?

-Were correct statistical analysis used to support conclusions?

-Are there concerns about ethical or regulatory requirements being met?

Reviewer #1: Looks great, the addition of mini-TAS methods is appreciated.

Reviewer #3: Thank you for addressing our comments, this made the paper richer and improved my understanding of the work.

**Results**

-Does the analysis presented match the analysis plan?

-Are the results clearly and completely presented?

-Are the figures (Tables, Images) of sufficient quality for clarity?

Reviewer #1: MUST UPDATE: Table 3 numbers do not match the text; Table 3 was updated to present % fail, and line 368 incorrectly reports "6.8% and 38.4% of replicates failing TAS, respectively" - these %, if you do the math, are for passing TAS. Recommend reviewing this complete sentence, and the Table 3 numbers, to decide what you want to present here - it could be that the % from Table 3 should be input into this sentence (alternatively, the word failing should be replaced with passing - however the complete sentence implies failure proportions are meant to be presented). Also, comment that Table 3 updated to % fail while Table 4 still presents % pass - in the event the authors wish to be consistent.

Reviewer #3: The revisions addressed the comments well and improved the paper.

**Conclusions**

-Are the conclusions supported by the data presented?

-Are the limitations of analysis clearly described?

-Do the authors discuss how these data can be helpful to advance our understanding of the topic under study?

-Is public health relevance addressed?

Reviewer #3: Reviewers' concerns appear to have been well addressed.

**Editorial and Data Presentation Modifications?**

Reviewer #3: Double check wording on line 198.

Figure Files:

Data Requirements:

Reproducibility:

References

---

## [Editor Report · Decision Letter 2]

6 Jan 2022

Dear Dr Gass,

We are pleased to inform you that your manuscript 'Simulating the effect of evaluation unit size on eligibility to stop mass drug administration for lymphatic filariasis in Haiti' has been provisionally accepted for publication in PLOS Neglected Tropical Diseases.

Best regards,

Luc E. Coffeng, MD PhD

Guest Editor

Jennifer Keiser

Deputy Editor

---

## [Editor Report · Acceptance letter]

24 Jan 2022

Dear Dr Gass,

We are delighted to inform you that your manuscript, "Simulating the effect of evaluation unit size on eligibility to stop mass drug administration for lymphatic filariasis in Haiti," has been formally accepted for publication in PLOS Neglected Tropical Diseases.

Best regards,

Shaden Kamhawi

co-Editor-in-Chief

Paul Brindley

co-Editor-in-Chief
